# REINFORCEMENT LEARNING WITH EX-POST MAX-MIN FAIRNESS

## ABSTRACT

We consider reinforcement learning with vectorial rewards, where the agent receives a vector of $K \geq 2$ different types of rewards at each time step. The agent aims to maximize the minimum total reward among the $K$ reward types. Different from existing works that focus on maximizing the minimum expected total reward, i.e. *ex-ante max-min fairness*, we maximize the expected minimum total reward, i.e. *ex-post max-min fairness*. Through an example and numerical experiments, we show that the optimal policy for the former objective generally does not converge to optimality under the latter, even as the number of time steps $T$ grows. Our main contribution is a novel algorithm, Online-ReOpt, that achieves near-optimality under our objective, assuming an optimization oracle that returns a near-optimal policy given any scalar reward. The expected objective value under Online-ReOpt is shown to converge to the asymptotic optimum as $T$ increases. Finally, we propose offline variants to ease the burden of online computation in Online-ReOpt, and we propose generalizations from the max-min objective to concave utility maximization.

## 1 INTRODUCTION

The prevailing paradigm in reinforcement learning (RL) concerns the maximization of a *single scalar reward*. On one hand, optimizing a single scalar reward is sufficient for modeling simple tasks. On the other hand, in many complex tasks there are often multiple, potentially competing, rewards to be maximized. Expressing the objective function as a single linear combination of the rewards can be constraining and insufficiently expressive for the nature of these complex tasks. In addition, a suitable choice of the linear combination is often not clear a priori.

In this work, we consider the reinforcement learning with max-min fairness (RL-MMF) problem. The agent accumulates a vector of $K \geq 1$ time-average rewards $\bar{V}_{1:T} = (\bar{V}_{1:T,k})_{k=1}^{K} \in \mathbb{R}^{K}$ in $T$ time steps, and aims to maximize $\mathbb{E}[\min_{k \in \{1,...,K\}} \bar{V}_{1:T,k}]$. The maximization objective represents *ex-post* max-min fairness, in contrast to the objective of *ex-ante* max-min fairness by maximizing $\min_{k \in \{1,...,K\}} \mathbb{E}[\bar{V}_{1:T,k}]$.

Our main contributions are the design and analysis of the Online-ReOpt algorithm, which achieves near-optimality for the *ex-post* max-min fairness objective. More specifically, the objective under Online-ReOpt converges to the optimum as $T$ increases. Our algorithm design involves a novel adaptation of the multiplicative weight update method (Arora et al., 2012), in conjunction with a judiciously designed re-optimization schedule. The schedule ensures that the agent adapts his decision to the total vectorial reward collected at a current time point, while allowing enough time for the currently adopted policy to converge before switching to another policy.

En route, we highlight crucial differences between the *ex-ante* and *ex-post* max-min fairness objectives, by showing that an optimal algorithm for the former needs not converge to the optimality even when $T$ increases. Finally, our results are extended to the case of maximizing $\mathbb{E}[g(\bar{V}_{1:T})]$, where $g$ is a Lipschitz continuous and concave reward function.

## 2 RELATED WORKS

The Reinforcement Learning with Max-Min Fairness (RL-MMF) problem described is related to an emerging body of research on RL with *ex-ante* concave reward maximization. The class of *ex-ante* concave reward maximization problems include the maximization of $g(\mathbb{E}[\bar{V}_{1:T}])$, as well as its *ex-ante variants*, including the long term average variant $g(\mathbb{E}[\lim_{T\to\infty} \bar{V}_{1:T}])$ and its infinite horizon discounted reward variant. The function $g : \mathbb{R}^K \to \mathbb{R}$ is assumed to be concave.

The class of ex-ante concave reward maximization problems is studied by the following research works. Chow et al. (2017) study the case where $g$ is specialized to the Conditional Value-at-Risk objective. Hazan et al. (2019) study the case when $g$ models the entropy function over the probability distribution over the state space, in order to construct a policy which induces a distribution over the state space that is as close to the uniform distribution as possible. Miryoosefi et al. (2019) study the case of minimizing the distance between $\mathbb{E}[\bar{V}_{1:T}]$ and a target set in $\mathbb{R}^K$. Lee et al. (2019) study the objective of state marginal matching, which aims to make the state marginal distribution match a given target state distribution. Pareto optimality of $\mathbb{E}[\bar{V}_{1:T}]$ and its ex-ante variants are studied in (Mannor & Shimkin, 2004; Gábor et al., 1998; Barrett & Narayanan, 2008; Van Moffaert & Nowé, 2014). Lastly, a recent work Zahavy et al. (2021) provides a unifying framework that encompasses many of the previously mentioned works, by studying the problem of maximizing $g(\mathbb{E}[\bar{V}_{1:T}])$ and its ex-ante variants, where $g$ is concave and Lipschitz continuous. Our contributions, which concern the ex-post max-min fairness $\mathbb{E}[\min_{k\in\{1,\ldots,K\}} \bar{V}_{1:T,k}]$ and its generalization to the ex-post concave case, are crucially different from the body of works on the ex-ante case. The difference is further highlighted in the forthcoming Section 3.2.

Additionally, a body of works Altman (1999); Tessler et al. (2019); Le et al. (2019); Liu et al. (2020) study the setting where $g$ is a linear function, subject to the constraint that $\mathbb{E}[\bar{V}_{1:T}]$ (or its ex-ante variants) is contained in a convex feasible region, such as a polytope. There is another line of research works Tarbouriech & Lazaric (2019); Cheung (2019); Brantley et al. (2020) focusing on various online settings. The works Tarbouriech & Lazaric (2019); Cheung (2019) focus on the ex-post setting like ours, but they crucially assume that the underlying $g$ is smooth, which is not the case for our max-min objective nor the case of Lipschitz continuous concave functions. In addition, the optimality gap (quantified by the notion of regret) degrades linearly with the number of states, which makes their applications to large scale problems challenging. Brantley et al. (2020) focus on the ex-ante setting, different from our ex-post setting, and their optimality gap also degrades linearly with the number of states.

## 3 MODEL

**Set up.** An instance of the Reinforcement Learning with Max-Min Fairness (RL-MMF) problem is specified by the tuple $(\mathcal{S}, s_1, \mathcal{A}, T, \mathcal{O})$. The set $\mathcal{S}$ is a finite state space, and $s_1 \in \mathcal{S}$ is the initial state. In the collection $\mathcal{A} = \{\mathcal{A}_s\}_{s\in\mathcal{S}}$, the set $\mathcal{A}_s$ contains the actions that the agent can take when he is at state $s$. Each set $\mathcal{A}_s$ is finite. The quantity $T \in \mathbb{N}$ is the number of time steps.

When the agent takes action $a \in \mathcal{A}_s$ at state $s$, he receives the array of stochastic outcomes $(s', U(s,a))$, governed by the outcome distribution $\mathcal{O}(s,a)$. For brevity, we abbreviate the relationship as $(s', U(s,a)) \sim \mathcal{O}(s,a)$. The outcome $s' \in \mathcal{S}$ is the subsequent state he transits to. The outcome $U(s,a) = (U_k(s,a))_{k=1}^K$ is a random vector lying in $[-1,1]^K$ almost surely. The random variable $U_k(s,a)$ is the amount of type-$k$ stochastic reward the agent receives. We allow the random variables $s', U_1(s,a), \ldots U_K(s,a)$ to be arbitrarily correlated.

**Dynamics.** At time $t \in \{1, \ldots T\}$, the agent observes his current state $s_t$. Then, he selects an action $a_t \in \mathcal{A}_{s_t}$. After that, he receives the stochastic feedback $(s_{t+1}, V_t(s_t, a_t)) \sim \mathcal{O}(s_t, a_t)$. We denote $V_t(s_t, a_t) = (V_{t,k}(s_t, a_t))_{k=1}^K$, where $V_{t,k}(s_t, a_t)$ is the type-$k$ stochastic reward received at time $t$. The agent select the actions $\{a_t\}_{t=1}^T$ with a policy $\pi = \{\pi_t\}_{t=1}^T$, which is a collection of functions. For each $t$, the function $\pi_t$ inputs the history $H_{t-1} = \cup_{q=1}^{t-1}\{s_q, a_q, V_q(s_q, a_q)\}$ and the current state $\{s_t\}$, and outputs $a_t \in \mathcal{A}_{s_t}$. We use the notation $a_t^\pi$ to highlight that the action is chosen under policy $\pi$. A policy $\pi$ is stationary if for all $t, H_{t-1}, s_t$ it holds that $\pi_t(H_{t-1}, s_t) = \bar{\pi}(s_t)$ for some function $\bar{\pi}$, where $\bar{\pi}(s) \in \mathcal{A}_s$ for all $s$. With a slight abuse of notation, we identify a stationary policy with the function $\bar{\pi}$.

**Objective.** We denote $\bar{V}_{1:t}^\pi = \frac{1}{t}\sum_{q=1}^t V_q(s_q, a_q^\pi)$ as the time average vectorial reward during time 1 to $t$ under policy $\pi$. The agent's over-arching goal is to design a policy $\pi$ that maximizes $\mathbb{E}[g_{\min}(\bar{V}_{1:T}^\pi)]$, where $g_{\min} : \mathbb{R}^K \to \mathbb{R}$ is defined as $g_{\min}(v) = \min_{k\in\{1,\dots,K\}} v_k$. Denoting $\bar{V}_{1:T,k}^\pi$ as the $k$-th component of the vector $\bar{V}_{1:T}^\pi$, the value $g_{\min}(\bar{V}_{1:T}^\pi) = \min_k \bar{V}_{1:T,k}^\pi$ is the minimum time average reward, among the reward types $1,\dots,K$. The function $g_{\min}$ is concave, and is 1-Lipschitz w.r.t. $\|\cdot\|_\infty$ over the domain $\mathbb{R}^K$.

When $K = 1$, the RL-MMF problem reduces to the conventional RL problem with scalar reward maximization. The case of $K > 1$ is more subtle. Generally, the optimizing agent needs to focus on different reward types in different time steps, contingent upon the amounts of the different reward types at the current time step. Since the max-min fairness objective could lead to an intractable optimization problem, we aim to design a near-optimal policy for the RL-MMF problem.

### 3.1 REGRET

We quantify the near-optimality of a policy $\pi$ by the notion of regret, which is the difference between a benchmark $\mathrm{opt}(\mathsf{P}(g_{\min}))$ and the expected reward $\mathbb{E}[g_{\min}(\bar{V}_{1:T}^\pi)]$. Formally, the regret of a policy $\pi$ in a $T$ time step horizon is

$$\mathrm{Reg}(\pi, T) = \mathrm{opt}(\mathsf{P}(g_{\min})) - \mathbb{E}[g_{\min}(\bar{V}_{1:T}^\pi)]. \tag{1}$$

The benchmark $\mathrm{opt}(\mathsf{P}(g_{\min}))$ is a fluid approximation to the expected optimum. To define $\mathrm{opt}(\mathsf{P}(g_{\min}))$, we introduce the notation $p = \{p(s'|s,a)\}_{s\in\mathcal{S}, a\in\mathcal{A}_s}$, where $p(s'|s,a)$ is the probability of transiting to $s'$ from $s, a$. In addition, we introduce $v = \{v(s,a)\}_{s\in\mathcal{S}, a\in\mathcal{A}_s}$, where $v(s,a) = \mathbf{E}[U(s,a)]$ is the vector of the $K$ expected rewards. The benchmark $\mathrm{opt}(\mathsf{P}(g_{\min}))$ is the optimal value of the maximization problem $\mathsf{P}(g_{\min})$. For any $g : \mathbb{R}^K \to \mathbb{R}$, we define

$$\mathsf{P}(g): \quad \max_x \; g\left(\sum_{s\in\mathcal{S}, a\in\mathcal{A}_s} v(s,a)x(s,a)\right)$$

$$\text{s.t.} \sum_{a\in\mathcal{A}_s} x(s,a) = \sum_{s'\in\mathcal{S}, a'\in\mathcal{A}_{s'}} p(s|s',a')x(s',a') \quad \forall s\in\mathcal{S} \tag{2a}$$

$$\sum_{s\in\mathcal{S}, a\in\mathcal{A}_s} x(s,a) = 1 \tag{2b}$$

$$x(s,a) \geq 0 \qquad\qquad \forall s\in\mathcal{S}, a\in\mathcal{A}_s. \tag{2c}$$

The concave maximization problem $\mathsf{P}(g_{\min})$ serves as a fluid relaxation to RL-MMF. For each $s\in\mathcal{S}, a\in\mathcal{A}_s$, the variable $x(s,a)$ can be interpreted as the frequency of the agent visiting state $s$ and taking action $a$. The set of constraints (2a) stipulates that the rate of transiting out of a state $s$ is equal to the rate of transiting into the state $s$ for each $s\in\mathcal{S}$, while the sets of constraints (2b , 2c) require that $\{x(s,a)\}_{s\in\mathcal{S}, a\in\mathcal{A}_s}$ forms a probability distribution over the state-action pairs. Consequently, $\mathrm{opt}(\mathsf{P}(g_{\min}))$ is an asymptotic (in $T$) upper bound to the expected optimum.

Our goal is to design a policy $\pi$ such that its regret[1] $\mathrm{Reg}(T)$ satisfies

$$\mathrm{Reg}(T) = \mathrm{opt}(\mathsf{P}(g_{\min})) - \mathbb{E}[g_{\min}(\bar{V}_{1:T}^\pi)] \leq \frac{D}{T^\gamma} \tag{3}$$

holds for all initial state $s_1 \in \mathcal{S}$ and all $T \in \mathbb{N}$, with parameters $D, \gamma > 0$ independent of $T$. We assume the access to an *optimization oracle* $\Lambda$, which returns a near-optimal policy given any scalar reward. For $\vartheta \in \mathbb{R}^K$, define the linear function $g_\vartheta : \mathbb{R}^K \to \mathbb{R}$ as $g_\vartheta(w) = \vartheta^\top w = \sum_{k=1}^K \vartheta_k w_k$. The oracle $\Lambda$ inputs $\vartheta \in \mathbb{R}^K$, and outputs a policy $\pi$ satisfying

$$\mathrm{opt}(\mathsf{P}(g_\vartheta)) - \mathbb{E}[g_\vartheta(\bar{V}_{1:T}^\pi)] = \mathrm{opt}(\mathsf{P}(g_\vartheta)) - \mathbb{E}[\vartheta^\top \bar{V}_{1:T}^\pi] \leq \frac{D_{\mathrm{lin}}}{T^\beta} \tag{4}$$

for all initial state $s_1 \in \mathcal{S}$ and all $T \in \mathbb{N}$, with parameters $D_{\mathrm{lin}}, \beta > 0$ independent of $T$. By assuming $\beta > 0$, we are assuming that the output policy $\pi$ is near-optimal, in the sense that the difference $\mathrm{opt}(\mathsf{P}(g_\vartheta)) - \mathbb{E}[\vartheta^\top \bar{V}_{1:T}^\pi]$ converges to 0 as $T$ tends to the infinity. A higher $\beta$ signifies a

---

[1]We omit the notation with $\pi$ for brevity sake

faster convergence, representing a higher degree of near-optimality. We refer to $\vartheta$ as a scalarization of $v$, with the resulting scalarized reward being $\vartheta^\top v(s, a)$ for each $s, a$.

Our algorithmic frameworks involve invoking $\Lambda$ as a sub-routine on different $\vartheta$'s. In other words, we assume an algorithmic sub-routine that solves the underlying RL problem with scalar reward (the case of $K = 1$), and delivers an algorithm that ensures max-min fairness (the case of $K \geq 1$). Finally, while the main text focuses on $g_{\min}$, our algorithm design and analysis can be generalized to the case of concave $g$, as detailed in Appendix C.

## 3.2 Comparison between maximizing $\mathbb{E}[g_{\text{MIN}}(\bar{V}_{1:T}^\pi)]$ and $g_{\text{MIN}}(\mathbb{E}[\bar{V}_{1:T}^\pi])$

Before introducing our algorithms, we illustrate the difference between the objectives of maximizing $\mathbb{E}[g_{\min}(\bar{V}_{1:T}^\pi)]$ and $g_{\min}(\mathbb{E}[\bar{V}_{1:T}^\pi])$ by the deterministic instance in Figure 1, with initial state $s_1 = s^o$.

The figure depicts an instance with $K = 2$. An arc represents an action that leads to a transition from its tail to its head. For example, the arc from $s^o$ to $s^\ell$ represents the action $a^{o\ell}$, with $p(s^\ell \mid s^o, a^{o\ell}) = 1$. Likewise, the loop at $s^\ell$ represents the action $a^{\ell\ell}$ with $p(s^\ell \mid s^\ell, a^{\ell\ell}) = 1$. Each arc is labeled with its vectorial reward, which is deterministic. For example, with certainty we have $V(s^o, a^{o\ell}) = \binom{0}{0}$ and $V(s^\ell, a^{\ell\ell}) = \binom{0}{1}$.

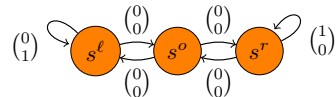

Figure 1: States and actions are represented by circles and arcs.

Consider two stationary policies $\pi^\ell, \pi^r$, defined as $\pi^\ell(s^r) = a^{ro}, \pi^\ell(s^o) = a^{o\ell}, \pi^\ell(s^\ell) = a^{\ell\ell}$ and $\pi^r(s^r) = a^{rr}, \pi^r(s^o) = a^{or}, \pi^r(s^\ell) = a^{\ell o}$. The policy $\pi^\ell$ always seeks to transit to $s^\ell$, and then loop at $s^\ell$ indefinitely, likewise for $\pi^r$. With certainty, $\bar{V}_{1:T}^{\pi^\ell} = \binom{0}{1-1/T}, \bar{V}_{1:T}^{\pi^r} = \binom{1-1/T}{0}$.

The objective $g_{\min}(\mathbb{E}[\bar{V}_{1:T}^\pi])$ is maximized by choosing $\pi_{\text{ran}}$ uniformly at random from the collection $\{\pi^\ell, \pi^r\}$. We have $\mathbb{E}[\bar{V}_{1:T}^{\pi_{\text{ran}}}] = \binom{1/2-1/(2T)}{1/2-1/(2T)}$, leading to the optimal value of $1/2 - 1/(2T)$. More generally, existing research focuses on maximizing $g(\mathbb{E}[\bar{V}_{1:T}^\pi])$ for certain concave $g$, and the related objectives of maximizing $g(\lim_{T\to\infty} \mathbb{E}[\bar{V}_{1:T}^\pi])$ or $g(\mathbb{E}[\sum_{t=1}^\infty \alpha^t V_t(s_t, a_t^\pi)])$, where $\alpha \in (0, 1)$ is the discount factor. In these research works, a near-optimal policy $\pi$ is constructed by first generating a collection $\Pi$ of stationary policies, then sampling $\pi$ uniformly at random from $\Pi$.

Interestingly, $\pi_{\text{ran}}$ is sub-optimal for maximizing $\mathbb{E}[g_{\min}(\bar{V}_{1:T}^\pi)]$. Indeed, $\Pr(\bar{V}_{1:T}^{\pi_{\text{ran}}} = \binom{0}{1-1/T}) = \Pr(\bar{V}_{1:T}^{\pi_{\text{ran}}} = \binom{1-1/T}{0}) = 1/2$, so we have $\mathbb{E}[g_{\min}(\bar{V}_{1:T}^{\pi_{\text{ran}}})] = 0$ for all $T$. Now, consider the deterministic policy $\pi_{\text{sw}}$, which first follows $\pi^\ell$ for the first $\lfloor T/2 \rfloor$ time steps, then follows $\pi^r$ in the remaining $\lceil T/2 \rceil$ time steps. We have $\bar{V}_{1:T,k}^{\pi_{\text{sw}}} \geq 1/2 - 2/T$ for each $k \in \{1, 2\}$, meaning that $g_{\min}(\bar{V}_{1:T}^{\pi_{\text{sw}}}) \geq 1/2 - 2/T$. Note that $g_{\min}(\mathbb{E}[\bar{V}_{1:T}^{\pi_{\text{sw}}}]) \geq g_{\min}(\mathbb{E}[\bar{V}_{1:T}^{\pi_{\text{ran}}}]) - 2/T$, so the policy $\pi_{\text{sw}}$ is also near-optimal for maximizing $g_{\min}(\mathbb{E}[\bar{V}_{1:T}^\pi])$.

Altogether, an optimal policy for maximizing $g_{\min}(\mathbb{E}[\bar{V}_{1:T}^\pi])$ can be far from optimal for maximizing $\mathbb{E}[g_{\min}(\bar{V}_{1:T}^\pi)]$. In addition, for the latter objective, it is intuitive to imitate $\pi_{\text{sw}}$, which is to partition the horizon into episodes and run a suitable stationary policy during each episode. A weakness to $\pi_{\text{sw}}$ is that its partitioning requires the knowledge on $T$. While our algorithm follows the intuition to imitate $\pi_{\text{sw}}$, we propose an alternate partitioning that allows does not require $T$ as an input.

## 4 Online-ReOpt Algorithm for RL-MMF

We propose the Online-ReOpt algorithm, displayed in Algorithm 1. The algorithm runs in episodes. An episode $m \in \{1, 2, \ldots\}$ starts at time $\tau(m)$ (defined in Line 2), and ends at time $\tau(m + 1) - 1$. Before the start of episode $m$, the algorithm computes the scalarization $\vartheta_{\tau(m)}$ based on the Multiplicative Weight Update (MWU), which we detail later. Then, the algorithm invokes the optimization oracle $\Lambda$, which returns a policy $\pi_m$ that is near-optimal for the MDP with scalar rewards $r_m = \{r_m(s, a)\}_{s,a}$, where $r_m(s, a) = \vartheta_{\tau(m)}^\top v(s, a)$. Note that we only assume a black-box access to $\Lambda$, and the parameters $D_{\text{lin}}, \beta$ do not need to be input to the Algorithm. Finally, the algorithm runs policy $\pi_m$ during episode $m$. The Online Re-Opt algorithm is an *anytime* algorithm, since it does not require $T$ as an input. Rather, it requires knowing $T$ only during the terminal time step $T$. To complete the description of the algorithm, we provide the details about the scalarization.

---

**Algorithm 1** Online-ReOpt for $g_{\min}$

---
1: Inputs: Optimization oracle $\Lambda$.
2: Set $\tau(m) = \lfloor m^{3/2} \rfloor$ for $m \in \mathbb{N}$.
3: **for** Episode $m = 1, 2, \ldots$ **do**
4:     Define $\vartheta_{\tau(m)}$ according to (5).
5:     Compute policy $\pi_m \leftarrow \Lambda(\vartheta_{\tau(m)})$.
6:     **for** Time $t = \tau(m), \ldots, \tau(m+1) - 1$ **do**
7:         Choose action $a_t = \pi_m(s_t)$.
8:         Observe the outcomes $V_t(s_t, a_t)$ and the next state $s_{t+1}$.
9:         **if** $t = T$ **then**
10:           Break the **for** loops and terminate the algorithm.
11:         **end if**
12:     **end for**
13: **end for**

---

**Scalarization by MWU.** At a time step $t$, we define the scalarization $\vartheta_t = \{\vartheta_{t,k}\}_{k=1}^K$ as

$$\vartheta_{t,k} = \frac{\exp\left[-\eta_{t-1} \sum_{j=1}^{t-1} V_{j,k}(s_j, a_j)\right]}{\sum_{\kappa=1}^K \exp\left[-\eta_{t-1} \sum_{j=1}^{t-1} V_{j,\kappa}(s_j, a_j)\right]}, \tag{5}$$

where

$$\eta_{t-1} = \frac{\sqrt{\log K}}{\max\{(t-1)^{2/3}, 1\}}. \tag{6}$$

In particular, at the start of each episode $m$, we apply (5) with $t = \tau(m)$ in Line 4. For the case $m = 1$, we have $\vartheta_{\tau(1),k} = 1/K$ for all $k \in \{1, \ldots, K\}$, meaning that all reward types are assigned with the same weight at the beginning. The exponent $2/3$ in the learning rate $\eta_{\tau(m)-1}$ in (5) is different from the conventional choice of $1/2$ (Arora et al., 2012). Our exponent is chosen for optimizing the resulting regret bound from our forthcoming analysis. We follow the approach in Chapter 7 in (Orabona, 2019) to define a time-varying learning rate.

The scalarization $\vartheta_t$ by (5) promotes max-min fairness. Consider two reward types $k, k'$ with $\sum_{q=1}^{t-1} V_{q,k}(s_q, a_q) > \sum_{q=1}^{t-1} V_{q,k'}(s_q, a_q)$. We have $\vartheta_{t,k'} > \vartheta_{t,k}$, meaning that a higher weight is assigned to reward type $k'$ than type $k$. This implies that there is a higher emphasis on increasing the type-$k'$ reward, which is in shortage as compared to type $k$, than the type-$k$ reward. Hence, max-min fairness is promoted.

## 4.1 THEORETICAL GUARANTEES

We provide the following theoretical guarantee for Online-ReOpt.

**Theorem 1** *Consider the RL-MMF problem. Online-ReOpt, displayed in Algorithm 1, satisfies*

$$Reg(T) \leq \frac{114\sqrt{\log K}}{T^{1/3}} + \frac{144 D_{lin}}{T^{\beta/3}}, \tag{7}$$

*where $D_{lin}, \beta$ are parameters related to the optimization oracle $\Lambda$.*

Theorem 1 is a generalization result, in the sense that it generalizes the ability of achieving near-optimality for the case of $K = 1$ to the case of $K \geq 1$. Indeed, as long as $\beta > 0$, meaning that the regret of the optimization oracle $\Lambda$ diminishes with a growing $T$ on any RL with scalar reward problem, the regret bound (7) in Theorem 1 also tends to zero as $T$ increases.

Theorem 1 is proved in Appendix section C.3. The first regret term in (7) arises from two sources: (a) the regret of the MWU algorithm, (b) the update delay on the scalarization due to the episodic structure. To elaborate on (b), consider a time step $t$ in episode $m$. Recall that the scalarization $\vartheta_t$ by (5) promotes max-min fairness, and ideally we should have employed the policy returned by $\Lambda(\vartheta_t)$ at time $t$. In contrast, in Online-ReOpt the action $a_t$ is determined by $\pi_m$, the output of $\Lambda(\vartheta_{\tau(m)})$.

Item (b) accounts for the regret due to using $\vartheta_{\tau(m)}$ rather then $\vartheta_t$. We crucially use the fact that $\vartheta_t$ are slowly changing in $t$ so that the resulting regret is still diminishing with $t$.

The second term in (7) is due to the regret of the optimization oracle $\Lambda$. The exponent $\beta/3$ in the term is less than the exponent $\beta$ in (4), as each policy $\pi_m$ is run for only $\tau(m+1) - \tau(m) = O(\sqrt{\tau(m)})$ many time steps. Our design of $\{\tau(m)\}_{m=1}^M$ allows a shorter time frame for $\pi_m$ to converge to its expected reward, as compared to running a policy for $T$ steps in (4). When we increase the episode length $\tau(m+1) - \tau(m)$, the regret due to (b) increases, while the regret due to the second term in (7) decreases, and vice versa. Our design of $\{\tau(m)\}_{m=1}^M$ strikes an optimal (in terms of our analysis) balance between these two sources of regret.

The regret bound (7) does not feature a direct dependence on the sizes of the state and action spaces. The dependence on the hardness of the underlying MDP is only reflected through the parameters $D_{\text{lin}}, \beta$. Therefore, apart from the deterioration of the exponent $\beta$ to $\beta/3$ and the first term in (7), our algorithm does not introduce any overhead in the generalization from the case of $K = 1$ to the case of $K \geq 1$. Improving the exponent $\beta/3$ in (7) is an interesting open question. Finally, Theorem 1 is generalized to the case of maximizing a concave utility objective $\mathbb{E}[g(\bar{V}_{1:T}^\pi)]$, where $g$ is Lipschitz continuous and concave. We detail the generalization in the model, algorithm and theoretical results in Appendix C.1.

## 4.2 OFFLINE VARIANTS TO ONLINE-REOPT

While the Online-ReOpt Algorithm achieves near-optimality, the efficiency of its implementation could be hindered by the need of online computation in Line 5 in Algorithm 1. Indeed, in order to compute $\pi_m$, the agent has to input the optimization oracle $\Lambda$ and the scalarization $\vartheta_{\tau(m)}$, which is only known at the end of time step $\tau(m) - 1$. In the case when the optimization oracle involves heavy computation, for example training deep neural networks, such online computation may not be realistic.

In this section, we propose Offline-ReOpt, which is a variant of Online-ReOpt that does not require invoking $\Lambda$ during the horizon. The Offline-ReOpt is obtained from the Online-ReOpt by modifying two lines in Algorithm 1, as enumerated below. The full algorithm of Offline-ReOpt is provided in Appendix section A.1.

1. Replace the input of $\Lambda$ in Line 1 with the input of the policy family $\Pi = \{(\vartheta, \pi^{(\vartheta)})\}_{\vartheta \in \Omega}$. The index set $\Omega$ is a finite subset of $\{\vartheta \in \mathbb{R}^K : \|\vartheta\|_1 = 1, \vartheta \geq 0\}$, the collection of all possible scalarizations. For each $\vartheta \in \Omega$, the policy $\pi^{(\vartheta)}$ is the output of $\Lambda(\vartheta)$.

2. Replace the online computation in Line 5 with these two lines:

   (a) Identify $\tilde{\vartheta}_{\tau(m)} \in \Omega$ that achieves $\min_{\vartheta \in \Omega} \left\| \vartheta - \vartheta_{\tau(m)} \right\|_1$.

   (b) Select policy $\pi_m = \pi^{(\tilde{\vartheta}_{\tau(m)})}$.

In item (1), all the policies in $\Pi$ are computed *before* the execution of the algorithm, unlike the case in Online-ReOpt. Consequently, in item (2), the selection of policy $\pi_m$ does not require invoking the optimization oracle $\Lambda$.

The main idea behind item (2) is that, in the case when the desired scalarization $\vartheta_{\tau(m)}$ does not lie in $\Omega$, we chooses the surrogate scalarization $\tilde{\vartheta}_{\tau(m)}$ that is closest to $\vartheta_{\tau(m)}$, so that the resulting policy $\pi^{(\tilde{\vartheta}_{\tau(m)})}$ will be a reasonable approximation to the desired policy $\pi^{(\vartheta_{\tau(m)})}$.

In order for the surrogate $\tilde{\vartheta}_{\tau(m)}$ to be close to $\vartheta_{\tau(m)}$, it is desirable for the finite index set $\Omega$ to be so diverse that every scalarization $\vartheta_{\tau(m)}$ would be in a close neighborhood of a scalarization in $\Omega$. We propose two families of $\Omega$ for the desired diversification. The first is the **random point family**, detailed in Appendix section A.2. The family is constructed by sampling random points in the domain $\{\vartheta \in \mathbb{R}^K : \|\vartheta\|_1 = 1, \theta \geq 0\}$ of all possible scalarizations. The second is the **imitation based family**, also detailed in Appendix section A.2. The family is constructed by first running Online-ReOpt multiple times, then collecting the scalarizations and the corresponding policies generated.

## 5 EXPERIMENTS

We evaluate our proposed algorithms and benchmark algorithms in a controlled queueing system involving vectorial rewards. For each of the algorithms, we first run the algorithm for $Z_{\mathrm{po}} = Z_{\mathrm{an}} \times \Xi$ independent trials, resulting in the $Z_{\mathrm{po}}$ average vectorial rewards[2] $\{\bar{V}_{1:T}^{(z_{\mathrm{an}},\xi)}\}_{1 \leq z_{\mathrm{an}} \leq Z_{\mathrm{an}}, 1 \leq \xi \leq \Xi}$. We plot the following three quantities against $T$:

- **Ex-post Fairness:** $\bar{\bar{\Psi}} = \frac{1}{Z_{\mathrm{an}}} \frac{1}{\Xi} \sum_{z_{\mathrm{an}}=1}^{Z_{\mathrm{an}}} \sum_{\xi=1}^{\Xi} g_{\min}\left(\bar{V}_{1:T}^{(z_{\mathrm{an}},\xi)}\right)$, an estimate to $\mathbb{E}[g_{\min}(\bar{V}_{1:T})]$.

- **Ex-ante Fairness:** $\bar{\Gamma} = \frac{1}{Z_{\mathrm{an}}} \sum_{z_{\mathrm{an}}=1}^{Z_{\mathrm{an}}} g_{\min}\left(\frac{1}{\Xi} \sum_{\xi=1}^{\Xi} \bar{V}_{1:T}^{(z_{\mathrm{an}},\xi)}\right)$, an estimate to $g_{\min}(\mathbb{E}[\bar{V}_{1:T}])$.

- **Type $k$ rewards for each** $1 \leq k \leq K$**:** $\bar{\bar{\Phi}}_k = \frac{1}{Z_{\mathrm{an}}} \frac{1}{\Xi} \sum_{z_{\mathrm{an}}=1}^{Z_{\mathrm{an}}} \sum_{\xi=1}^{\Xi} \bar{V}_{1:T,k}^{(z_{\mathrm{an}},\xi)}$, an estimate to $\mathbb{E}[\bar{V}_{1:T,k}]$.

We define the upper and lower error bars respectively as the 75 and 25-percentiles of the data, see Appendix section B.1 for details. For the forthcoming discussions, we denote $\mathbf{e}_k$ as the $k$-th standard basis vector for $k \in \{1, \ldots K\}$ in $\mathbb{R}^K$. In addition, we denote $\mathbf{1}_K, \mathbf{0}_K$ as the all one vector and the all zero vector in $\mathbb{R}^K$.

### 5.1 QUEUING NETWORK

Queuing problems are studied extensively due to their relevance in fields such as manufacturing and in communication systems. In our evaluation, we focus on a discrete-time queuing system. The queuing network that we have tested our algorithms on, consisting of two servers and four queues arranged in a bidirectional fashion, has been previously studied in works by Rybko & Stolyar (1992), Kumar & Seidman (1990), Chen & Meyn (1998), de Farias & Van Roy (2003) and Banijamali et al. (2019). This network is shown in Figure 2.

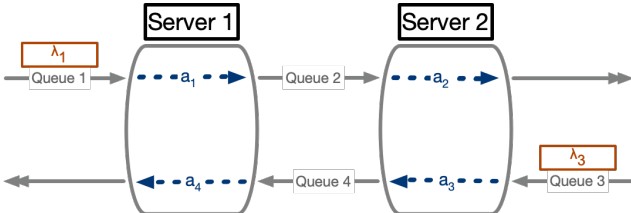

Figure 2: A bi-directional four-queue network

There are two servers in the system. Server 1 only serves Queue 1 *or* Queue 4 with service rates $\mu_1 = 0.3$ and $\mu_4 = 0.3$ respectively, and where Server 2 similarly only serves Queue 2 *or* Queue 3 at the rates of $\mu_2 = 0.3$ and $\mu_3 = 0.3$. Arrivals occur at a rate of $\lambda_1 = 0.2$ and $\lambda_2 = 0.2$ at Queues 1 and 3 respectively. An arrival that gets served at Queue 1 by Server 1 then progresses to Queue 2, and only leaves the system after it has been served by Server 2. Likewise, arrivals at Queue 3 have to be served by Server 2, before moving on to Queue 4 to be served by Server 1 in order to leave the system. Each queue $i$ has a maximum length of $L_i = 9$, and a customer is rejected at a queue if the queue is at its full capacity. Conversely, an empty queue remains at length 0 even if an action is taken to serve that queue.

The state of the system is thus defined by the vector $\boldsymbol{x}_t = (x_{t,1}, x_{t,2}, x_{t,3}, x_{t,4})$ whereby $x_{t,i}$ represents the length of the queue $i$ at time $t$. At each time step $t$, a decision has to be made by each server to serve *only one* or *neither* of its queues, which we can represent by a 4-component vector $\boldsymbol{a}_t = (a_{t,1}, a_{t,2}, a_{t,3}, a_{t,4}) \in \{0,1\}^4$, where $a_{t,i} = 1$ indicates the decision to serve Queue $i$ at time $t$, and $a_{t,i} = 0$ otherwise. Note that the condition of being able to only serve *one* queue at each server naturally imposes the constraints $a_{t,1} + a_{t,4} \leq 1$ and $a_{t,2} + a_{t,3} \leq 1$ at each $t$, meaning that $\mathcal{A}_s = \{a \in \{0,1\}^4 : a_1 + a_4 \leq 1, a_2 + a_3 \leq 1\}$ for each $s$.

---

[2]To avoid clutter, we omit the upper-script for the algorithm.

The transition dynamics for the system can then defined by the following equation when $0 < x_{t,i} < L_i$, where $\boldsymbol{e}_i$ refers to the basis vector in $\mathbb{R}^4$:

$$
\boldsymbol{x}_{t+1} = \begin{cases}
\boldsymbol{x}_t + \boldsymbol{e}_1 & \text{with probability } \lambda_1 \\
\boldsymbol{x}_t + \boldsymbol{e}_3 & \text{with probability } \lambda_2 \\
\boldsymbol{x}_t + \boldsymbol{e}_2 - \boldsymbol{e}_1 & \text{with probability } \mu_1 a_1 \\
\boldsymbol{x}_t - \boldsymbol{e}_2 & \text{with probability } \mu_2 a_2 \\
\boldsymbol{x}_t + \boldsymbol{e}_4 - \boldsymbol{e}_3 & \text{with probability } \mu_3 a_3 \\
\boldsymbol{x}_t - \boldsymbol{e}_4 & \text{with probability } \mu_4 a_4 \\
\boldsymbol{x}_t & \text{otherwise}
\end{cases}
\tag{8}
$$

We define the type-$i$ reward at time $t$ as $V_{t,i}(\boldsymbol{x}, \boldsymbol{a}) = 1 - \frac{x_{t,i}}{L_i}$, for $i \in \{1, \dots, 4\}$. Recall that $x_{t,i}$ is the queue length of Queue $i$ at time $t$. The reward $V_{t,i}(\boldsymbol{x}, \boldsymbol{a})$ is equal to 1 if Queue $i$ is empty, and the reward $V_{t,i}(\boldsymbol{x}, \boldsymbol{a})$ decreases linearly with the length of Queue $i$ at time $t$. In particular, $V_{t,i}(\boldsymbol{x}, \boldsymbol{a}) = 0$ if Queue $i$ is full. Altogether, the agent's reward for Queue $i$ at time $t$ positively correlates with the degree of idleness of the Queue. The maximization of $g_{\min}(\bar{V}_{1:T}) = \min_{1 \le i \le 4} \bar{V}_{1:T,i}$ is equivalent to the minimization of time-average queue lengths among all queues, hence enforcing all queues to be stable simultaneously.

### 5.1.1 SIMULATION RESULTS

In our simulation, we evaluate 5 algorithms. Three of them are our proposed algorithms, namely Online-ReOpt, Offline-ReOpt with Random Point Family and Offline-ReOpt with Imitation based family. The other two are existing baselines. The Meta Algorihtm by Zahavy et al. (2021) is the state-of-the-art for maximizing $g_{\min}(\mathbb{E}[\bar{V}_{1:T}])$, while the longer queue first heuristic is a well-established algorithm in the queuing theory literature. As the name suggests, each server serves the longer of the two queues at each time round. We ran each of the algorithms with the following parameters: $T = 100000$, $Z_{an} = 10$ and $\Xi = 100$, meaning $Z_{po} = Z_{an} \times \Xi = 1000$.[3] All of the algorithms employ the same optimization oracle $\Lambda$, with the same hyper-parameters and architecture, a double deep Q-learning network algorithm (Double DQN) by Hasselt et al. (2016).

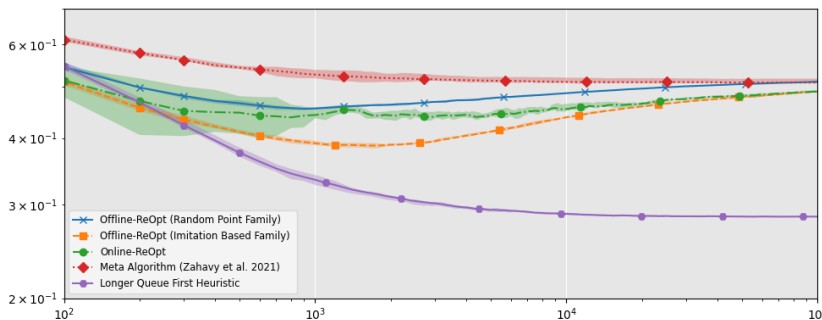

Figure 3: Ex-ante Fairness of various algorithms in a queuing network

Figure 3 plots the quantity $\bar{\bar{\Psi}}$ against $T$ under the 5 algorithms. Notice in Figure 3 how the Offline and Online-ReOpt algorithms, as well as the Meta Algorithm by Zahavy et al. (2021) perform similarly well in terms of *ex-ante* fairness. Among them, the Meta Algorithm has the best performance, since the Re-Optimization schedule in our proposed algorithms compromises the *ex-ante* fairness objective. All algorithms demonstrate converging behavior, in the sense that the error bars diminishes as $T$ grows.

---

[3]Except for Online-ReOpt, where we set $\Xi = 5$ since running an online algorithm for 1000 trials is not as practical as running an offline algorithm, which only needs to be trained once.

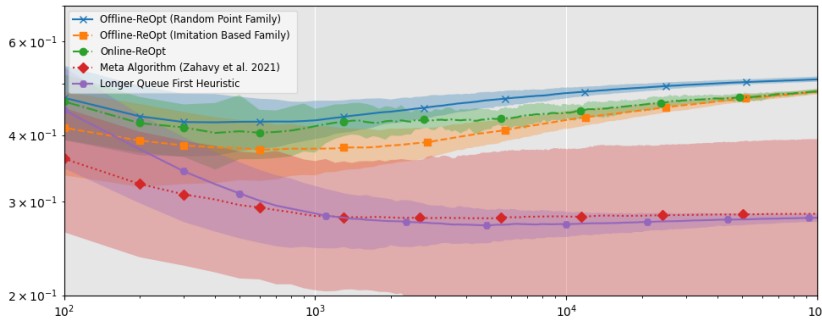

Figure 4: Ex-post Fairness of various algorithms in a queuing network

Figure 4 plots the quantity $\bar{\Gamma}$ against $T$ under the 5 algorithms. In terms of *ex-post* fairness, the Offline and Online-ReOpt algorithms perform significantly better than Meta Algorithm and the Longer Queue First Heuristic. The sub-optimality of the Meta Algorithm corroborates with Section 3.2 that policies designed for the *ex-ante* fairness objective could be far from optimal for the *ex-post* fairness objective. While the Meta Algorithm has a similar performance to the Longer Queue First heuristic, the former has a significantly wider error bar than the latter, meaning that the latter is more stable.

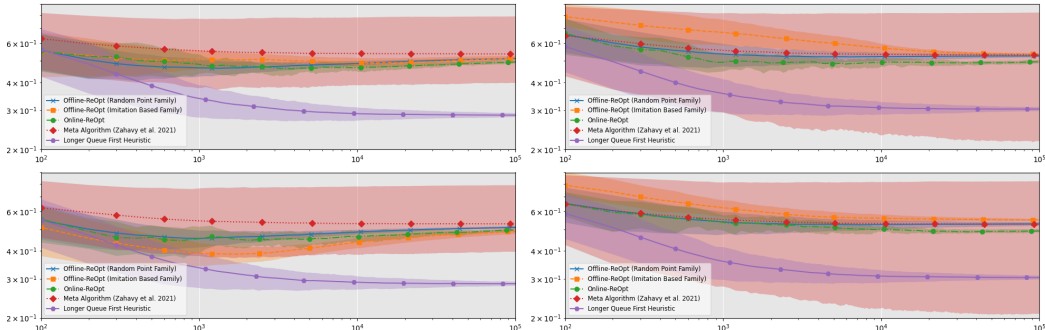

Figure 5: Type $k$-rewards for $1 \le k \le K$ of various algorithms in a queuing network. Figure is read from left to right, top to bottom.

Figure 5 plots $\Phi_i$ against $T$ for $i \in \{1, \ldots, 4\}$. In a nutshell, the plotted lines explain the trends in Figure 3, while the error bars shed light on the trends in Figure 4. Firstly, the plotted lines indicate that the Meta Algorithm has the highest (or close to the highest) individual average reward $\Phi_i$ for each queue, signifying that the Meta Algorithm has the highest $\mathbb{E}[\bar{V}_{1:T,i}]$ for each $i \in \{1, \ldots, 4\}$. This explains the superiority of the Meta Algorithm shown in Figure 3.

When we focus on the error bars, the plots in Figure 5 tell a different story. Notably, the error bars for the Meta Algorithm is significantly wider than others, meaning that the $Z_{\mathrm{po}}$ trials of the Meta Algorithm have significantly different results from one another.[4] When we unpack the summands in $\Phi_1, \ldots, \Phi_4$ and compute the minimum reward in each trial, it results in Figure 4, which is vastly different from Figure 3, signifying the ex-ante and ex-post objectives are fundamentally different.

As a final remark, our numerical experiments do not imply that the Longer Queue Heuristic is a worse algorithm than the other 4 algorithms. Indeed, the Longer Queue Heuristic does not require the knowledge of $\lambda_1, \lambda_2, \mu_1, \ldots, \mu_4$, whereas the other 4 algorithms crucially uses these parameters for generating their policies. In addition, the Longer Queue Heuristic is computationally much less onerous than the others. Finally, the Longer Queue Heuristic demonstrates converging behaviors in all the plots, in the sense that the error bars diminish when $T$ increases.

---

[4]It is helpful to revisit Section 3.2, where $Z_{\mathrm{po}}$ trials would result in $\approx Z_{\mathrm{po}}/2$ outcomes of $\binom{0}{1-1/T}$ and $\approx Z_{\mathrm{po}}/2$ outcomes of $\binom{1-1/T}{0}$.

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
