# OpenReview forum: "Reinforcement Learning with Ex-Post Max-Min Fairness"
_ICLR.cc/2022/Conference — ICLR 2022 Submitted_

### Official Review · Reviewer_gzxW · 2021-10-31

**Correctness:** 3
**Technical Novelty And Significance:** 2
**Empirical Novelty And Significance:** 2
**Recommendation:** 3
**Confidence:** 3

**Main Review:**

1. Some important related papers need reference: https://arxiv.org/pdf/1909.02940.pdf which seems to solve similar problem.
2. Note that even though the E[min] and min(E) are shown to be different in an example, the gap is O(1/T), and not really relevant to the problem of regret. Thus, I believe that the algorithm of ex-ante will give the ex-post result with similar regret.
3. Based on the above, it is unclear why the above paper approach where MDP is estimated using Dirichlet/optimistic sampling and a model based optimization is applied do not work in this setup? As T goes large, E[min] and min(E) will be similar and have gap of O(1/\sqrt{T}) and can be quantified giving an additional regret term, but other than that, most analysis will be re-used.
4. The novelty in the approach needs better discussions.
5. The regret bounds are sub-optimal. Given that oracle for linear reward problem will have \beta = 1/2, the regret is T^{5/6} which is very large when the model-based algorithms can achieve T^{1/2}.

**Summary Of The Paper:**

This paper considers ex-post max-min fairness, where an RL algorithm is provided with provable guarantees. The difference between ex-ante and ex-post are explained.

**Summary Of The Review:**

The key issue in the paper is the in-efficient regret bound, and the approach novelty needs better explanations.

---

> ### Author Response · Authors · 2021-11-17
> **Response to Reviewer gzxW**
>
> Please see the following on our response to the main review.
>
> 1. We will include the related reference (Agarwal et al.) in our revision. Our work is crucially different:
> -  Our algorithm works on a strictly broader set of instances than Agarwal et al., who crucially assume $p(s' | s, a)>0$ with probability 1 for all states $s', s$ and all actions $a$ under the Bayesian assumption of Dirichlet priors. This assumption is stronger than the uni-chain assumption (see Section 8.3 in Puterman 94), which says that for any states $s, s'$ and any stationary policy, the agent reaches $s'$ starting from $s$ under $\pi$ with finitely many state transitions in expectation. In comparison, our assumption is weaker than the assumption of communicating MDPs, which says that for any states $s, s'$, there exists a stationary policy such that the agent reaches $s'$ starting from $s$ under $\pi$ with finitely many state transitions in expectation.
>
> 1 (con'td). Their assumption presents 2 major limitation:
>
> - Unichain is hardly realistic, since any MDP with an action a such that $p(s | s, a) = 1$ would not be unichain. Such null actions are common in MDP models where the agent needs an option to stay put. The queueing instance we simulate is also not unichain.
>
> - Their algorithm is not applicable in Figure 1, while ours is. In their Algorithm 1 Line 16, they define a stationary policy $\pi(a | s) = \frac{x^*(s, a)}{\sum_{a} x^*(s, a)}$, where $x^*$ is an optimal solution to $\textsf{P}(g_\text{min})$. Under Figure 1, we have $x^*(s^r, a^{rr}) = x^*(s^\ell .  a^{\ell}) = 1/2$, and $x^*(s, a) = 0$ for other state action pairs $s, a$. Thus, at $s^\ell$, we have  $\pi(a^{\ell\ell} | s^\ell) =1$, meaning the agent loop at $s^\ell$ with probability 1. Likewise $\pi(a^{rr} | s^r) =1$. The true problem is with what to do at $s^o$. We see that $\sum_{a} x^*(s^o, a)$, so $\pi( a| s^o)$ is ill-defined. It is tempting to break tie by uniform sampling, i.e. $\pi(a^{0\ell}|s^o) = \pi(a^{0r}|s^o) = 1/2$, but then $\pi$ will be reduced to $\pi^\text{ran}$, which is shown to suffer $\text{Reg}(T) = \Omega(1)$ in Section 3.2. Alternatively, we seek a different route by imitating $\pi^\text{sw}$, which is non-stationary, different from $\pi^\text{ran}$.
>
> - In addition, as shown in Appendix C, our results hold for any Lipschitz continuous objective function, while (Agarwal et al. 2021) requires the objective function to be component-wise monotone (see their Assumption 3).
>
> - Agarwal et al's Algorithm 1 Line 16 requires solving the convex program $\textsf{P}(g_{\text{min}})$ (which is similar to Altman 1999), which is intractable in most reinforcement learning settings that involve large state or action spaces, since the feasible region has $O(|\mathcal{S}|)$ many constraints. In comparison, we provide a computationally more efficient approach, as our algorithm only requires an RL algorithm that optimizes a scalar reward, and such algorithms are well developed in the past decades.
>
> (2, 3) The claim about the gap between E[min] and min(E) being $O(1/T)$ is not true. Consider $\pi_{\text{ran}}$ in Section 3.2, we in fact have E[min] $=1/2 - 1/(2T)$, but min(E) = 0. Indeed, we have $\bar{V_{1:T}}$ is equal to $\binom{0}{1 - 1/T}$  with probability $1/2$ and equal to $\binom{1 - 1/T}{0}$  with probability $1/2$. Consequently, $E[\bar{V_{1:T}}] = \binom{1/2 - 1/(2T)}{1/2 - 1/(2T)}$, but $\Pr(\min_{k\in \{1, 2}} \bar{V_{1:T, k}} = 0) = 1$. Therefore, the gap does not diminish with $T$, and we cannot hope to use existing algorithms for $min(E)$ to handle $E(min)$.
>
> 4. The novelty of the paper lies on crucial difference between the objectives of E[min] and min(E). Please see our response to Reviewer bwVS for the detailed discussion.
>
> 5. While there could be potential for improvement, it is crucial to observe that the state-of-the-art algorithms do not even converge to the optimum (ie $\text{Reg}(T) = \Omega(1)$ does not diminish as $T$ grows), as we showed in Section 3.2 and also in our simulations. Our work is the first to achieve $\text{Reg}(T)$ diminishing with $T$ for the fairness objective beyond unichain (The unichain case is done by Altman 1999). Moreover, when the underlying MDP is communicating, in our optimization oracle assumption (4) we can set $D_\text{lin} = 2D$ where $D$ is the diameter of the MDP (see Theorem 38.2 and Lemma 38.3 in the "Bandit Algorithms" book: https://tor-lattimore.com/downloads/book/book.pdf), and $\beta = 0$, which results in $\text{Reg}(T) = O(1/T^{1/3})$.

---

### Official Review · Reviewer_oTCf · 2021-11-02

**Correctness:** 4
**Technical Novelty And Significance:** 2
**Empirical Novelty And Significance:** 2
**Recommendation:** 5
**Confidence:** 3

**Main Review:**

First, the paper is well-written and I really enjoy reading the work.
The paper proposes a new objective and an asymptotic optimal algorithm. The presentation is clear and intuitive, and the analysis is also very clean.

I mainly have two concerns:
(i) The motivation for the ex-post objective: The paper provides an example (Section 3.2) to illustrate the difference between ex-post and ex-ante policies. I like the example, but I still don't feel convinced about the necessity of the ex-post objective. Say, if we adopt a linear objective with equal weight or a CVaR objective to the two rewards for this example, would we end up with the same optimal policy as the ex-post? I think it would be great if the author(s) can construct an example, for which the optimal policy of the ex-post objective cannot be recovered by other existing objective functions, like the linear, ex-ante, CVaR, etc. In this sense, it means the ex-post objective provides us with a different class of risk-sensitive/fair policies.
Also, from a practical perspective, the numerical experiments should compare the time spent by the two classes of customers in the system, rather than the waiting time for each of the four queues. For instance, customer type 1 may care more about the sum of (rather than each of) the waiting times before Queue 1 and Queue 2. And for fairness purposes, we might want to max-min {Q1 Waiting + Q2 Waiting, Q3 Waiting + Q4 Waiting}. I am not sure if this would complicate the problem setup significantly. Please correct me if I misunderstood the setup here.

(ii) Knowledge of the underlying dynamics (reward and transition): The oracle used by the algorithm requires the knowledge of the underlying reward and transition, at least some estimates of them. Strictly speaking, due to this requirement, I wouldn't call the setup of the paper "RL" but more of an MDP or stochastic dynamic programming. It seems to me that this requirement is necessary for both the algorithm and the analysis. I wonder whether asymptotic optimality can still be obtained if we don't have this knowledge. For example, if we replace the true probabilities/rewards in the oracle as their empirical estimates based on the samples observed so far in the online algorithm, my hunch is that the performance guarantee will no longer hold.

**Summary Of The Paper:**

The paper considers the ex-post max-min objective for the vector reward RL problem. The paper provides a simple example to illustrate the difference between ex-ante and ex-post objectives. An MWU-based algorithm is proposed to solve the problem with a provable regret guarantee, where the algorithm resorts to an approximately optimal policy oracle episodically. The algorithm also features a variant that can fully rely on offline solutions. Numerical experiments on a classic queue control problem demonstrate the performance of the algorithm against two existing benchmarks.

**Summary Of The Review:**

Overall, as mentioned above, I think the paper is very elegant. I would look forward to hearing the author(s) comments on my two concerns above.

---

> ### Author Response · Authors · 2021-11-17
> **Response to Reviewer oTCf**
>
> Motivation for the ex-post objective. Our ex-post objective can be interpreted as requiring max-min fairness on a single trajectory. Indeed, $g_\text{max}$ is applied on the average $\bar{V_{1:T}}$ of a single run. In comparison, the ex-ante objective can be interpreted as requiring max-min fairness on the average of multiple trajectory. It is because $g_\text{min}$ is applied on the expectation $\mathbb{E}[\bar{V_{1:T}}]$, and the distance between $\mathbb{E}[\bar{V_{1:T}}]$ and the average of $M$ independent trials $\bar{V_{1:T}}(1), ... , \bar{V_{1:T}}(M)$ shrinks as $M$ grows. As we saw in Figure 1, when $M=1$, the distance does not shrinks with $T$.
>
> In Figure 1, our algorithm is able to visit each $s^\ell, s^r$ with frequency $1/2$ in a single $T$-step horizon. In contrast, with algorithms by Zahavy et al., they require to reposition the agent at $s^o$ and run $T$ time steps. Only after repeating this for $M$ time, then we can observe the convergence of the visitation frequencies to 1/2. That being said, without the repositioning their algorithm cannot achieve $\text{Reg}(T) = o(1)$, unlike Online-ReOpt. By contrast, our algorithm teaches to the agent to transit to $s^o$ at appropriate times to achieve fairness in one single trajectory.
>
> Difference from other formulations. Equal Weight: Consider Figure 1. If we put equal weights on $s^\ell,s^\r$ (for example, set $\vartheta = \binom{1}{1}$), then the scalarized objective is optimized by $\pi^\ell$, $\pi^r$, as well as the randomized policy $\pi$ where $\Pr(\pi = \pi^\ell) = p = 1 - \Pr(\pi = \pi^r)$ for $p \in (0, 1)$ (this are all the optimal policies). None of them achieves $\text{Reg}(T) = o(1)$, since
> $\bar{V}_{1:T}$ would realize as either $\binom{0}{1-1/T}$ or $\binom{1 - 1/T}{0}$, and $g_\text{min}(\binom{0}{1-1/T}) = g_\text{min}(\binom{0}{1-1/T}) = 0$.
>
> Ex-ante: The ex-ante and ex-post objectives are not mutually exclusive. Note that $\text{opt}(\textsf{P}(g_\text{min})$ is also an upper bound to the optimum under the ex-ante objective (see Zahavy et al.). Therefore, by concavity of $g_\text{min}$, our algorithm has both its ex-ante regret $\text{opt}(\textsf{P}(g_\text{min}) - g_\text{min}(E[\bar{V_{1:T}}])$  and ex-post regret $\text{opt}(\textsf{P}(g_\text{min}) - E[g_\text{min}(\bar{V_{1:T}})]$  diminishing in $T$. By contrast, the algorithms by Zahavy et al. (as well as Hazan et al. and Miryoosefi et al.) only guarantee the ex-ante regret to diminish with $T$, and we have seen in Section 3.2 that their ex-post regret does not.
>
> CVaR: We believe that the CVaR formulation (for example in Chow et al. 2017) is not compatible with the max-min fairness objective. Indeed, such formulations involve the CVaR of the cumulative scalar reward in the objective or constraints, with one reward type per object/constraint. In contrast, the max-min fairness require comparing rewards of different types. In order to incoprate max-min fairness with CVaR, the ``$\alpha$'' would need to depend the cumulative reward of other types, and it is not clear how $\alpha$ should be chosen.
>
> To elaborate further, the existing approaches are insufficient for the ex-post objective, since these approaches always return a stationary policy. Any (randomized) stationary policy $\pi$ fails to achieve a $T$-diminishing ex-post regret in Figure 1. Denote $s^\pi_t$ as the random state visited at time $t$ under $\pi$. If $\lim_{T\rightarrow \infty }\frac{1}{T}\sum^T_{t=1}\mathsf{1}(s^\pi_t = s^o) > 0$, then clearly $\text{Reg}(T) = \Omega(1)$ since it requires $\lim_{T\rightarrow \infty }\frac{1}{T}\sum^T_{t=1}\mathsf{1}(s^\pi_t = s^\ell)  = \lim_{T\rightarrow \infty }\sum^T_{t=1} \frac{1}{T}\mathsf{1}(s^\pi_t = s^r) =1/2$ to ensure $\text{Reg}(T)$ shrinks with $T$. Otherwise, we have $\lim_{T\rightarrow \infty } \sum^T_{t=1} \frac{1}{T}\mathsf{1}(s^\pi_t = s^o) = 0$, and $\pi$ is a mixture of $\pi^\ell, \pi^r$ (see Section 3.2), which won't achieve $\text{Reg}(T) = o(1)$ as discussed before.
>
> Queueing. Yes, the customer level fairness is also a desirable metric. It will not complicate the set up, since we only need to define $K=2$ and to redefine the vectorial rewards as suggested. Numerical experiments are being conducted.
>
> Knowledge of model. Yes, if the model is latent and needs to be learnt online, our framework will not directly apply, and we suppose that learning algorithm such as Jaksch et al. will need to be incorporated. Our main motivation is to capitalize the wealth of RL algorithms on scalar rewards, in the sense that our Online-ReOpt is a blackbox transformation from scalar reward RL algorithms to max-min fairness RL algorithms, and in our analysis we show how the error in the optimization oracle (equation (4)) affect the regret for the max-min fairness objective. One of the main observation is that max-min fairness does not incur overhead loss due to state or action space sizes, while we expect online learning on latent model would incur such loss.

---

### Official Review · Reviewer_SQsM · 2021-11-02

**Correctness:** 4
**Technical Novelty And Significance:** 2
**Empirical Novelty And Significance:** 2
**Recommendation:** 3
**Confidence:** 3

**Main Review:**

Strengths:
1. The authors provide both theorectical analysis and empirical experiments. In the analysis, they use multiplicative weight update to ensure the max-min fairness. In the numerical experiments, the proposed algorithm outperforms previous algorithms significantly.


Weaknesses:
1. The intuition of ex-post max-min fairness is not clear. The authors try to show difference between ex-ante max-min fairness and ex-post max-min fairness. However, the two problems are almost the same for weak communicating MDPs. In weak communicating MDPs,  for any policy $\pi$, by running $\pi$ for $T$ steps, $||\bar{V}_{1:T}-E[\bar{V}^{\pi}]||$ is bounded by $\tilde{O}(K\sqrt{D/T})$ with high probability, where $V^{\pi}$ is the expected average reward of $\pi$ and $D$ is the diameter. The proposed counter example is a non-communicating MDP, which is less important in both theorectical analysis and practical problems (in general, we deal with non-communicating MDPs by dividing it into several weak communicating parts).

2. The requirement of the proposed algorithm is too strong. Online-ReOpt needs to take an oracle to compute a near-optimal policy for any  scalar reward as a subrountine. In general, by online learning it cost $O(SAD/\epsilon^2)$ to learn an $\epsilon$-optimal policy in the worst case. Although it is possible to add a subroutine to learn such a policy, it would worse the final regret considering the current regret bound is quite poor.

**Summary Of The Paper:**

The paper considers RL problem with a vectorial reward of $K$ dimension. The authors consider to maximize the minimal value function $\mathbb{E}[g_{min}(\bar{V}_{1:T})]$ and achieve a regret bound of $O(\sqrt{logK}T^{\frac{2}{3}})$ given an oracle to solve the RL problem with scalar reward. The authors also discuss the offline variant of the proposed algorithm. Finally, the authors verify their algorithm with a queuing system.

**Summary Of The Review:**

Overall the paper is well written and easy to read. However, given the concerns above, I tend to reject it.

---

> ### Author Response · Authors · 2021-11-17
> **Response to Reviewer SQsM**
>
> About the two weaknesses:
>
> 1. The problems of ex-ante and ex-post max-min fairness are different when the underlying MDP is communicating.
>
> a. The instance in Figure 1 is a communicating MDP. Recall that an MDP is communicating if for any two states $s, s'$, there exists a policy that start at $s$, and reaches $s'$ with finitely many state transitions in expectation. In Figure 1, the agent can travel from a state to another  with at most two transitions with certainty. For example, starting from $s^\ell$, the agent can first follows the upper arc $a^{o\ell}$ to $s^o$, and then follows the upper arc $a^{or}$ to $s^r$.
>
> b. The claim "in a (weakly)-communicating MDP, for running $\pi$ for $T$ steps, $||\bar{V}^\pi_{1:T} - \mathbb{E}[\bar{V^\pi_{1:T}}]|| \leq \tilde{O}(K\sqrt{D/T})$" is not true. Consider the communicating MDP Figure 1, and the policy $\pi = \pi^{\text{ran}}$. Our discussions in Section 3.2 indicates that $||\bar{V}^\pi_{1:T} - \mathbb{E}[\bar{V^\pi_{1:T}}]|| = (\frac{1}{2} - \frac{1}{2T}) || \binom{1}{-1} ||$ with certainty. Indeed, we have demonstrated that $\Pr(\bar{V_{1:T}} = \binom{0}{1 - 1/T}) = \Pr(\bar{V_{1:T}} = \binom{1 - 1/T}{0}) = 1/2$, which implies $ \mathbb{E}[\bar{V}^{\pi_\text{ran}}_{1:T}] = \binom{1/2 - 1/(2T)}{1/2 - 1/(2T)}$. As long as the norm satisfies $|| \binom{1}{-1} || \neq 0$, quantity $(\frac{1}{2} - \frac{1}{2T}) || \binom{1}{-1} ||$ does not diminish even when $T\rightarrow \infty$.
>
> Altogether, we saw that $\pi^\text{ran}$, while being optimal for ex-ante fairness, does not achieve ex-post fairness. This is all because $||\bar{V}^\pi_{1:T} - \mathbb{E}[\bar{V^\pi_{1:T}}]||$ generally does not diminish with $T$. (If the claim were true, we would have agreed with your concern.) Our core contribution is to recognize this subtlety, and design a different algorithms to get around it.
>
> In passing, the algorithms by Zahavy et al. (and by Hazan et al., Miryoosefi et al), which are specifically for the ex-ante case of maximizing $g(E[\bar{V}])$, will not overcome the issue in Figure 1. Indeed, these algorithms first involve generating a family $\Pi$ of stationary policies by gradient descent, then randomly sampling a policy $\pi$ from $\Pi$ and then running $\pi$ for the $T$ round horizon. In Figure 1, those algorithms would produce a family $\Pi$, which is a multi-set consisting of $\pi^\ell, \pi^r$. Assuming there are $N_\ell, N_r$ copies of $\pi^\ell, \pi^r$ in $\Pi$, we would have $\Pr(\pi = \pi^\ell) = \frac{N^\ell}{N^\ell + N^r} = 1 - \Pr(\pi = \pi^r)$, and consequently $\Pr(\bar{V_{1:T}} = \binom{0}{1 - 1/T}) = \frac{N^\ell}{N^\ell + N^r}
>  = 1-\Pr(\bar{V_{1:T}} = \binom{1 - 1/T}{0})$. With their algorithms, $\frac{N^\ell}{N^\ell + N^r} $ converges to $1/2$ as the number of training episodes increases.
>
> This motivates us to consider a different algorithm design that involves partitioning the time horizon into epochs and run different stationary policies for different epochs (largely inspired by $\pi^\text{sw}$ in Section 3.2), different from prior works who always run the same stationary policy on the whole horizon.
>
> 2. We can weaken the optimization oracle assumption, by replacing $D_\text{lin} / T^{\beta}$, the right hand side of (4) on the optimization oracle assumption, with $\epsilon >0$. This means that the oracle is only approximately optimal in the sense that there is an optimality gap of $\epsilon$ no matter how large $T$ is. As a result, the regret bound in (7) will be changed by replacing the second term $144 D_\text{lin} / T^{\beta / 3}$ with $\epsilon$.
>
> Indeed, with the change in assumption, we only need to modify the analysis on the term $(\ddagger)$ (see Appendix Section C.3.2 in the supplementary), and right hand side of (33) would be replaced by $\text{opt}(\textsf{P}(g_{-\theta_{\tau(m)}})) - \epsilon$. Consequently, line (34) will be replaced by $\mathbb{E}[\ldots]  - \frac{1}{T}\sum^M_{m=1} (\tau(m+1) - \tau(m))\epsilon = \mathbb{E}[\ldots]  - \epsilon$, and the analysis on $\mathbb{E}[\ldots]$ follows exactly on the last page in the appendix.
>
> The reason why we see a deterioration with our initial assumption in (4) is that, the assumption (4) is an assumption on convergence rate (parameterized by $\beta$), and we show that the resulting rate in the Theorem could be worsened to $\beta/3$. But with a fixed error  bound $\epsilon$ that does not converge as $T$ grows, the bound stays the same under our framework.

---

> > ### Comment · Reviewer_SQsM · 2021-11-22
> > **Response to the rebuttal**
> >
> > Thank the authors for the feedback.
> > 1. About the motivation of ex-post max-min fairness: Sorry for the mistake in the claim you mentioned above. The mistake could be fixed by replacing $D$ by $D_{\pi}$, where $D_{\pi}$ is the diameter following $\pi$. It is possible that $D_{\pi}$ is very large so that the two terms differ. However, for any fixed policy $\pi$ ($\pi$ could be non-stationary), $E[\bar{V}^{\pi}]$ could be approximated (with error $T^{-1/3}$) by $\bar{V_{1:T}}$  following a non-stationary policy $\pi'$ defined as below.
> > Let $N=$ and $T_{k}=\frac{kT}{N}$. Given $\pi$, let $D_k$ denote the distribution of state at time $T_k$ for $1\leq k \leq N$. Then we let
> >  $\pi'$ runs following $\pi$ for the first $T_1 = \frac{T}{N}$ steps and then taking some policy $\pi_1$ to reach $D_1$. Since the diameter of the MDP is $D$, it is possible to reach $D_1$ in $D$ steps in expectation. Then we let $\pi'$ runs following $\pi$ given the initial distribution $D_1$ for the next $\frac{T}{N}$ steps. By doing this for $k=1,2,...,\ldots$, we can show that $||\bar{V_{1:T}}- E[\bar{V}^{\pi}]||$ is bounded by $O(\frac{t_1+t_2+...+t_N}{T}+\frac{1}{\sqrt{N}})$, where $t_i$ is the number of steps used to achieve $D_i$.
> > Here the first term is due to the steps using to reach the ideal distribution, and the second term is by the gap between empirical values and expectation. Taking expectation for $t_i$ and noting that $E[t_i]\leq D$, we have that  $||\bar{V_{1:T}}- E[\bar{V}^{\pi}]||$ is bounded by $O(DN/T+1/\sqrt{N})$. Choosing $N = T^{2/3}$, the proof is finished.
> >
> >
> > 2.  It is possible to choose $\epsilon$ adaptively according to $T$ with the doubling trick. I think it is worth trying.

---

> > > ### Author Response · Authors · 2021-11-22
> > > **Response to Reviewer SQsM**
> > >
> > > Thanks for the thoughtful suggestions, and let us input some additional comments:
> > >
> > > 1. The reviewer's suggestion is in alignment with our design idea that, in order to have $\bar{V}$ converge to $E[\bar{V}^\pi]$ for a certain policy $\pi$, we generally have to follow another non-stationary policy $\pi'$ different from $\pi$. A crucial question is: How (computationally) hard is it to construct $\pi'$?
> > >
> > > On one hand, the reviewer's suggestion could provide a way to achieve the convergence in the case of communicating MDPs. On the other hand, the suggestion is computationally intensive. Indeed, the suggestion requires constructing a policy that travels from state $s$ to state $s'$ for every $s, s'\in S$, which is not scalable in large state instances. For example, suppose that $D_k$ is the uniform distribution over the state space. For every state $s\in S$, the agent has to construct  a policy $\pi_s$ that travel from $s_{T_k}$, the state at time $T_k$ which is random, to $s$, and then to take a uniform distribution over $\{\pi_s\}_{s\in S}$ in order to achieve $D_k$.  In addition, the computation of $D_k$ also appears to be rather onerous.
> > >
> > > In comparison, with our approach, we only require solving $\lceil T^{2/3} \rceil$ many scalarized MDP problems with our Online-ReOpt for achieving a $O(1/T^{1/3})$ when the agent's optimization oracle is optimal for communicating MDPs (meaning that $\beta=0$ in equation 4). This assumption would be needed with the reviewer's approach for solving the problem, where we suppose that $\pi$ is an optimal policy for the ex-ante problem. What we show is a blackbox transformation from an oracle that solves the scalarized problem to an algorithm that achieves ex-post max-min fairness. In particular, we show that the In fact, our results are more general in the sense that we quantify how the error in the optimization oracle affects the final regret bound. In addition, with our offline variants, the computationally burden also does not increase with the size of the state space.
> > >
> > > Lastly, while we have been discussing on the special case of communicating MDPs, we remark that our optimization oracle assumption is more general than assuming that the underlying MDP to be communicating. In particular, in our algorithm, there is no requirement for the agent to walk from one specific state to another.
> > >
> > > 2. We note that our optimization oracle assumption is a blackbox assumption. While we assume that (4) holds for certain error parameters $D_\text{lin}, \beta$ (or the error parameter $\epsilon$ suggested by the reviewers), these parameters are not known and they are only used in our analysis. We only assume the possession of the oracle that inputs a scalarization and outputs a policy. Consequently, these error parameters do not appear in Algorithm 1. Nevertheless, we agree that it would be an interesting direction to tune the error parameters in the underlying optimization oracle, under the assumption that the oracle is specified to be a certain class of algorithms such as A3C.

---

### Official Review · Reviewer_bwVS · 2021-11-02

**Correctness:** 3
**Technical Novelty And Significance:** 3
**Empirical Novelty And Significance:** 3
**Recommendation:** 5
**Confidence:** 3

**Main Review:**

**Strength:** The problem studied is interesting and well-motivated. The paper also clearly illustrated the difference between ex-ante and ex-post optimizations. The main algorithm presented has a theoretical regret bound. The paper is clearly written overall.

**Weakness:** The main techniques seem to rely on the work of Zahavy et al. (Reward is Enough for Convex MDPs), especially the generalization to concave minimization. Hence, the work looks somewhat incremental from a technical point of view. The authors may want to stress the difference between their work and that of Zahavy et al in the paper and highlight the original contribution. That said, the application of these techniques to the maximin problem is still very interesting.

There is one technical question, which might be critical and I hope the authors could clarify: I do not see how Aogrithm 1 avoids getting stuck in a sink state that is undesirable for some reward types. For example, in Figure 1, if there is an additional state $s'$, connected by an arc from $s^o$ and with a self loop generating the same rewards as the self loop at $s^l$; nevertheless, from $s'$ no other states can be reached. In this case, a policy that always seeks to transit to $s'$ and loops there indefinitely is also optimal w.r.t. the second dimension of the reward. Therefore, the oracle $\Lambda$ may well pick this policy in the first episode, but if this is the case, the agent would loop at $s'$ in the subsequent episodes, obtaining reward $0$ for the first dimension. I wonder how the regret could diminish in this case? Is some assumption of ergodicity relevant here? I don't find any in the paper though.

**Minor comments and suggestions:**

A typo: Right above Section 4: "...allows does not require..."

The figures in the experiment section could be arranged in a better way. They all look overly wide, and those in Figure 5 too small to read. If the space is a concern, I'd sugget putting Figures 3 and 4 side by side to save space for Figure 5.

**Summary Of The Paper:**

The paper studies an RL problem, where the rewards are given as a vector at each time step. The goal is to find a policy that balances the total rewards on different dimensions of the reward vector, i.e., one that maximizes the total reward of the worst dimension. More specifically, the paper takes an ex-post perspective and presents an online algorithm that achieves near optimality. An offline variant of this online algorithm is also proposed to alleviate the heavy cost of online computation in some applications. The authors also conducted experiments to evaluate the proposed algorithms.

**Summary Of The Review:**

The ex-post maximin fairness problem is interesting and well-motivated. The authors made an effort to develop an algorithm to solve the problem, which looks somewhat incremental from a technical point of view given an existing result by Zahavy et al. (Reward is Enough for Convex MDPs). There is a potential technical issue (see Main Review) which I hope the authors could clarify.

---

> ### Author Response · Authors · 2021-11-17
> **Response to Reviewer bwVS**
>
> Compared to Zahavy et al, our contributions involve the following novelty:
>
> 1. Performance guarantee. Our algorithm has both the ex-ante regret  $\text{opt}(\textsf{P}(g_\text{min})) - g_\text{min}(\mathbb{E}[ \bar{V_{1:T}} ])$  and ex-post regret  $\text{opt}(\textsf{P}(g_\text{min}))  - \mathbb{E}[g_\text{min}(\bar{V}_{1:T})]$ diminishing with $T$. Under While the ex-ante regret of the algorithm by Zahavy et al diminishes with $T$, its ex-post regret does not (as shown in Section 3.2). Note Zahavy et al also uses $\text{opt}(\textsf{P}(g_\text{min}))$ as the benchmark in their regret, so our comparison on the ex-ante regret is valid.
>
>
> 2. Algorithm design and analysis. Both Zahavy et al. and ours involves online mirror descent (OMD). However, the applications of OMD by Zahavy et al. and ours are fundamentally different. Zahavy et al. involves generating a collection $\Pi$ of stationary polices using OMD, and then sampling $\pi$ uniformly at random from $\Pi$, followed by applying $\pi$ in the $T$ time steps. In contrast, ours involves a suitable partition of the $T$-time-step horizon into episodes, and we only invokes the OMD at the start of each episodes. The episodes structure signifies a delay in applying the gradient descents, since after $\tau(m)$, we only update the gradient at time $\tau(m+1)$. This requires us to bound the regret due to delaying the gradients update (see the analysis in Appendix C.3.1), which differs from Zahavy et al. who do not delay the updates for the ex-ante objectives.
>
>
> 3. Necessity of a different design. The ex-post regret $\text{Reg}(T)$ of algorithm by Zahavy et al. 2021 does not diminish in the instance in Figure 1, as shown in Section 3.2. More generally, we cannot expect the approach by Zahavy et al. 2021, which involves randomly sampling a stationary policy from a suitably chosen family $\Pi$, to work. Indeed, for Figure 1, the approach by Zahavy et al. 2021 produces a randomized stationary policy, which is randomized over $\pi^\ell, \pi^r$. However, as hinted in Section 3.2, if we are to choose $\pi^\ell$ with probability (w.p.) $p$ and $\pi^r$ w.p. $1-p$, the resulting $\bar{V}_{1:T}$ will be $= \binom{0}{1-1/T}$ w.p. $p$ and $= \binom{1-1/T}{0}$ w.p. $1- p$, which still leads to $\text{Reg}(T) = \Omega(1)$.
>
> This motivates us to look for a different design, which essentially involves "bridging" $\pi_\text{sw}$ and $\pi_\text{ran}$. Subjectively, we find them rather different looking, but we constructed a hybrid (namely Online-ReOpt) that bridges the two and generalizable to a wider class of instances.
>
> Technical question: In the reviewer's instance, Reg(T) will not diminish as $T$ grows. Our results is not directly applicable in the instance, since the optimization oracle assumption (4) is violated. By specifying $\vartheta = \binom{1}{0}$ and initial state $s_1= s'$, we see that $\text{opt}(\textsf{P}(g_\vartheta))  = 1$, while $ \mathbb{E}[g_\vartheta(\bar{V_{1:T}})]  = 0$, since the agent is totally trapped (looped) at $s'$ and he can only earn a scalarized reward of 0. The failure of achieving $\text{Reg}(T) = o(1)$ is in fact largely because the benchmark $\text{opt}(\textsf{P}(g_\text{min})$ is equal to $1/2$ in the reviewer's instance. By contrast, since $s'$ is a trapping state with vectorial reward $\binom{0}{1}$, we have  $ \mathbb{E}[g_\text{min}(\bar{V}^{\pi}_{1:T})]  = 0$.
>
> From the above discussions, our assumption (4) does impose some sort of an ergodicity assumption, in the sense that the assumption could fail in a multi-chain (see Section 8.3 in Puterman 94) MDP, which is the reviewer's case. In comparison, assumption (4) is satisfied by any communicating MDP, with $D_\text{lin} = 2D$ where $D$ is the diameter of the MDP (see Theorem 38.2 and Lemma 38.3 in the "Bandit Algorithms" book: https://tor-lattimore.com/downloads/book/book.pdf), and $\beta = 0$.
>
> To optimize for the reviewer's instance, which is a non-communicating MDP, we can in fact first dividing it into several weak communicating parts (as alluded by Reviewer SQsM),  and then apply our algorithm on each, which would have satisfied our assumption (4).
>
> Lastly, we remark that such technical question does not arise in the ex-ante case of maximizing g(E($\bar{V}$)), but arises in our ex-post case. This further shows the difference between ex-ante and ex-post fairness.

---

### Decision · Program_Chairs · 2022-01-20

**Decision:**

Reject

**Comment:**

This paper studies an RL problem with vector rewards, where the goal is to maximize the expected minimum total reward (ex-post max-min fairness). This is different from prior works on a similar topic, where the goal is to maximize the minimum expected total reward (ex-ante max-min fairness). The authors propose an algorithm for solving the problem with $O(T^{2 / 3})$ regret and evaluate it.

This paper received two borderline reject and two reject reviews. The reviewers recognize the novelty of the objective. However, they are also concerned with its motivation and that the proposed algorithm relies on strong assumptions, such as that the used oracle knows the underlying reward and transition models, or at least has some estimate of them. At the end, the scores of this paper are not good enough for acceptance. Therefore, it is rejected.